

# Influencing factors and their interactions of water erosion based on yearly and monthly scale analysis: A case study in the Yellow River basin of China

Ting Hua[1,2], Wenwu Zhao[1,2], Yanxu Liu[1,2], and Yue Liu[1,2]

[1]State Key Laboratory of Earth Surface Processes and Resource Ecology, Faculty of Geographical Science, Beijing Normal University, Beijing 100875, China

[2]Institute of Land Surface System and Sustainable Development, Faculty of Geographical Science, Beijing Normal University, Beijing 100875, China

*Correspondence to*: Wenwu Zhao (zhaoww@bnu.edu.cn)

**Abstract.** In the Yellow River basin, soil erosion is a significant natural hazard problem, seriously hindering the sustainable development of society. An in-depth assessment of soil erosion and a quantitative identification of the influencing factors are important and fundamental for soil and water conservation. The RUSLE model and geographical detector method were applied to evaluate and identify the dominant factors and spatiotemporal variability in the Yellow River basin. We found that topographical factors such as slope and surface roughness were the dominant factors influencing the spatial distribution of soil erosion in the Yellow River basin, while rainfall and vegetation were as follows. In the period of low rainfall and vegetation coverage, the interaction of rainfall and slope can enhance their impact on the distribution of soil erosion, while the combination of vegetation and slope was the dominant interacting factor in other periods. The dominant driving factors of soil erosion variability were affected by changes in rainfall, but the contribution decreased. The spatial and temporal heterogeneity of soil erosion on a monthly scale was higher, and July had the highest amount of soil erosion with a multi-year average of 12.385 ton/(km²·a). The results provide a better understanding of the relationships between soil erosion and its latent factors in the Yellow River basin. Given the temporal and spatial heterogeneity effects of geographical conditions, especially at the basin scale, policy-makers should form a collaborative environmental governance framework to minimize the risk of soil erosion.

## 1 Introduction

Soil erosion has the potential to change soil structure and negatively affects soil fertility, land productivity, food security, biological diversity and the global carbon (C) cycle; additionally, soil erosion is likely the most dangerous form of soil degradation worldwide (Amundson et al., 2015; Van Oost et al., 2012; Alexandridis et al., 2015; Keesstra et al 2016; Lal, R., 2004). It is a global environmental and ecological issue that seriously hinders the sustainable development of society (Borrelli et al., 2017; Martinez-Casasnovas et al., 2016; Kefi et al., 2011). Although a large number of soil erosion assessments have been carried out on different spatial scales, the relationships between environmental factors and soil





erosion are not consistent among various research conditions. How to quantify the effect of
environmental factors on the distribution and variability of soil erosion, especially considering the
interaction of environmental factors, is still a question that must be answered by conducting multiple
analyses of regions that experience high soil erosion.

The identification of the mechanisms of soil erosion and factors affecting soil erosion is an

important basis for land use management and ecosystem government. Several studies have focused on
determining the driving forces affecting soil erosion, including precipitation, geomorphology, land use
type, vegetation, and soil physical properties (Vrieling, 2006; Zhou et al., 2008; Peng and Wang, 2012;
Gao and Wang., 2018; Beskow et al., 2009; Tian et al., 2009). The splashing function of raindrops and
the runoff generated by rainfall are the main driving factors of soil erosion. As the slope increases, the
amount of soil erosion and the rate of increase of soil erosion both increase. For vegetation, the vegetation
canopy can protect the surface soil from direct impact from raindrops and weaken runoff, thus eventually
reducing soil erosion. The Yellow River, especially the middle reaches located on the Loess Plateau, is
the region with the most serious soil erosion caused by water in the world (Liu and Liu, 2010; Sun et al.,
2014). The Chinese Government has undertaken numerous soil conservation projects in the Yellow River,
especially the Grain-for-Green Program that started in 1998, which has greatly improved the ecological
and environmental quality in this region and is expected to influence soil erosion (Gao et al., 2011; Fu et
al., 2011). Sun et al. explored the effects of rainfall, vegetation cover, land cover and topography on soil
erosion risk in the Loess Plateau (2013;2014).  Zhao et al. identified the risk of soil erosion in the middle
reaches of the Yellow River from 1978 to 2010 dynamically (2018). Du et al. assessed the risk caused
by water and wind in the watershed of the Ningxia-Inner Mongolia reach of the Yellow River (2016).

Previous studies have primarily been concerned with the identification and quantification of single

factors; however, research on the effects of multi-factor interactions on soil erosion is insufficient. The
variation in precipitation will influence the soil water content, further influence the development of
vegetation, and eventually decrease or accelerate erosion (Hou et al., 1996). In addition, the decreased
rainfall reduces the rainfall erosivity and eventually lowers the amount of soil erosion, but it may also
lower the density of vegetation cover due to insufficient water. Therefore, the relationships among
precipitation, vegetation, topography and erosion are uncertain due to their complex interactions, and





quantitative studies of their contributions and multiple interacting factors are important. These studies
are important and necessary for policy-makers to develop soil and water protection measures.

Large-scale soil erosion monitoring relies heavily on the development of models, and the Revised

Universal Soil Loss Equation (RUSLE) is the most widely applied empirical erosion model based on the
Universal Soil Loss Equation (USLE) (Wishmeier and Smith, 1978; Renard et al., 1997). Using the
detailed surface information provided by remote sensing, the RUSLE model has successfully been
applied to a variety of spatial scale assessments of soil erosion, from the plot scale to the global scale
(Thiam, 2003; Vrieling, 2006; Van der kniff, 1999; Van der kniff, 2000; Borrelli et al., 2013).
Specifically, for the RUSLE model, the soil erodibility (K factor) and topography (LS) factors are stable
over a long time period and are relatively independent of anthropogenic interventions. However, the
rainfall erodibility (R factor) and vegetation cover and management factor (C factor) are seasonally
variable. The C factor is the most adjustable factor based on land use management (Durán Zuazo and
Rodríguez Pleguezuelo, 2008; Maetens et al., 2012; Biddoccu et al., 2014; Eshel et al., 2015; Biddoccu
et al., 2016), with the highest amplitude of spatial and temporal variation among all the RUSLE factors
(Estrada-Carmona et al., 2016). Similar to the C factor, the contribution of the R factor is also the
amplitude of the spatial and temporal variation caused by the large variability in the monthly rainfall
under the context of climate change. Because of seasonal changes in these environmental factors, the
annual scales of soil erosion assessments often ignore more detailed fluctuations, and the effects of
factors related to soil erosion must also have the same seasonal effects. Furthermore, the focus of soil
and water conservation work is closely related to the seasonal fluctuation of soil erosion and its driving
factors. Compared to existing annual scale studies, more detailed time-scale soil erosion assessments are
urgently needed, which would help establish the effects and trends of various factors on soil erosion and
develop soil and water conservation strategies based on seasonal fluctuations.

The aim of this work is to study the dominant factors influencing soil erosion and temporal change

in the Yellow River basin of China. The specific objectives include the following: (1) obtain the
distribution and monthly variation of soil erosion in the Yellow River basin; (2) quantitatively identify
the dominant factors affecting the distribution pattern and variability of soil erosion on a yearly and
monthly scale.





## 2 Data and methods

### 2.1 Study area

The study area is the Yellow River basin. The Yellow River has a total length of 5,464 km and a drainage area of 795,000 $km^2$, accounting for 8.28% of China's land area (Figure 1). According to statistics from 1997, the population of the Yellow River basin was $1.07 \times 10^8$, accounting for 8.6% of the national population; additionally, the area of cultivated land in the Yellow River basin was $1.26 \times 10^7 km^2$, accounting for 13.3% of the country's cultivated land and making it an important agricultural development zone in China. However, soil erosion in the Yellow River basin, especially in the middle reaches of the Loess Plateau, has become an important environmental problem that hinders local agricultural and socio-economic development. Therefore, the soil and water conservation work in the Yellow River basin is of great significance to the sustainable development of the basin.

### 2.2 Data and processing

2.2.1 The RUSLE model

The soil erosion was estimated by the RUSLE model (Renard et al., 1997), which was revised based on the USLE model (Wishmeier and Smith, 1978). This model has been used to simulate and assess soil erosion worldwide using GIS and remote sensing tools. The equation is as follows:

$$A = R \times K \times LS \times C \times P, \tag{1}$$

where A is the soil erosion module, R is the rainfall erosivity factor, K is the soil erodibility factor, LS is the slope aspect factor, C is the land cover and management factor, and P is the conservation measure factor.

The R factor was computed using a diurnal rainfall model based on the Köppen climatic zone. The Yellow River basin contains 6 Köppen climatic zones: BS (arid and steppe), BW (arid and steppe), Cf (warm temperate and fully humid), Cw (warm temperate and winter dry), Dw (snow and dry winter) and Df (snow and fully humid). The specific R factor formula is as follows:

$$EI = \alpha P^\beta + \varepsilon, \tag{2}$$

where $P$ is the daily rainfall data, and the values of $\alpha$, $\beta$, and $\varepsilon$ depend on the climate region. The parameters are shown in Table S2. Rainfall data from 1995 to 2015 were acquired from the National



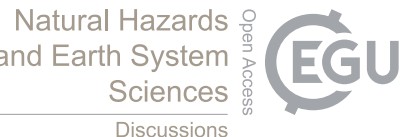

Meteorological Information Center (http://data.cma.cn/). A gridded rainfall erosivity dataset with a
spatial resolution of 1000 m at monthly and yearly scales was interpolated using ANUSPLIN 4.2
software (Hutchinson, 2001), with data from 240 meteorological stations in the Yellow River basin and
its surrounding areas.

We computed the soil erodibility (K factor) using the land erosion-productivity impact model (EPIC)

developed by Williams et al. (1990) as follows:
$K = \left[0.2 + 0.3e^{-0.0256SAN\left(1-\frac{SIL}{100}\right)}\right]\left(\frac{SIL}{CLA+SIL}\right)^{0.3}\left(1.0 - \frac{0.25C}{C+e^{3.72-2.95C}}\right)\left(1.0 - \frac{0.7SN_1}{SN_1+e^{-5.51+22.9SN_1}}\right),$     (3)
where SAN is the percent sand content, SIL is the percent silt content, CLA is the percent clay content,
C is the percent organic carbon content, and $SN_1$ = 1 – SAN/100.

Factors L and S were calculated based on the interaction of topography and flow accumulation.

Thus, the 90 m digital elevation model (DEM) dataset STRM3 DEM (http://srtm.csi.cgiar.org/) was used.
For S, the formula of McCool et al. (1987) was selected for slopes below 10º, and the formula of Liu et
al. (1994) was used for slopes above 10º. The specific formula is as follows:
$S = 10.8 \times \sin\theta + 0.03\ (\theta < 5°),$                                                    (4)
$S = 16.8 \times \sin\theta - 0.5\ (5° \leq \theta < 10°),$                                           (5)
$S = 21.9 \times \sin\theta - 0.96\ (10° \leq \theta),$                                               (6)
where  θ is the slope value.

The L factor was computed using the method developed by Liu et al. (2010), based on the

expression in Foster and Wischmeier (1974).
$L_i = \frac{\lambda_{out}^{m+1} - \lambda_{in}^{m+1}}{(\lambda_{out} - \lambda_{ini-1})22.13^m},$                                                          (7)
$m = \begin{cases} 0.2 & \theta \leq 0.5° \\ 0.3 & 0.5° < \theta \leq 1.5° \\ 0.4 & 1.5° < \theta \leq 3° \\ 0.5 & \theta > 3° \end{cases},$                                    (8)
where $L_i$ is the L factor of the $i$-th grid, $\lambda_{out}$ and $\lambda_{in}$ are the slope lengths of the exit and entrance,
respectively, and m is the slope length index.



The C factor is defined as the ratio of soil loss under the given vegetation cover to that which would
occur under continuously bare soil. The C factors were acquired from previous large-scale studies in
Europe (Van der kniff,1999,2000), and the detailed equation is as follows:
$C = \exp(-2(\text{NDVI}/(1 - \text{NDVI}))),$            (9)
where the NDVI is the normalized difference vegetation index. The NDVI images were acquired by the
Global Inventory Modelling and Mapping Studies (GIMMS) NDVI 3g V1.0, which has a 15-day spatial
resolution of 1/12 degrees that is available globally (https://ecocast.arc.nasa.gov/data/pub/gimms/3
g.v1/). Using the maximum value composite (MVC) method, we generated monthly NDVI data based
on two corresponding 15-day datasets and used the average of the generated monthly NDVI dataset to
obtain the annual NDVI dataset. P is the supporting practice. Due to the lack of data and the spatial
resolution of the research, this value was set to 1.
The Climate Change Initiative land cover (CCI LC) project developed by the European Space
Agency with a spatial resolution of 300 m was also used in this study. The temporal frame of analysis
included 20 years from 1995 to 2015, with particular attention to the five temporal nodes of 1995, 2000,
2005, 2010 and 2015.
2.2.2 Geographical detector
The geographical detector is a spatial variance analysis method developed to detect the
heterogeneity of an event and assess the relationship between the event and its potential risk factors,
including environmental and anthropogenic factors (Wang et al., 2010). The core idea is based on the
assumption that if an independent variable $X$ has an important influence on a dependent variable $Y$, then
the spatial distributions of the independent variable $X$ should have similarities (Wang et al., 2012, Wang
et al., 2017). The proportion of the spatial distribution of dependent variable $Y$ that can be explained by
independent variable $X$ is measured by the power of determinant ($q$ value). The calculation is as follows:
$q = 1 - \frac{1}{N\sigma^2}\sum_{Z=1}^{L} N_Z \sigma_Z^2,$           (10)
$\sigma_z^2 = \frac{1}{N_z - 1}\sum_{i=1}^{N_z}\left(Y_{z,i} - \overline{Y_Z}\right)^2,$         (11)
$\sigma_z^2 = \frac{1}{N - 1}\sum_{j=1}^{N}\left(Y_j - \overline{Y}\right)^2,$          (12)



where $\sigma^2$ is the variance of $Y$ in the region, $\sigma_2$ is the variance in zone $Z$ divided by $X$, $N$ is the number
of sample units in the region, $N_Z$ is the number of sample units in zone $Z$, and $L$ is the number of
zones. $Y_{z,i}$ and $Y_j$ are the values of $Y$ in the $i$-th sample units of zone $Z$ and the $j$-th sample unit of the
entire region, respectively.

Two modules provided by a geographical detector, a factor detector module and an interaction

detector module are used in this study. The factor detector module probes the extent to which factor $X$
(independent variable) explains the spatial differentiation of attribute $Y$ (dependent variable), and the $q$
value of the interaction between two influencing factors was calculated using the interaction detector
module. The input dataset (independent variable $X$) that a geographical detector requires must be
discretized, such as a land use dataset and a continuous value dataset, such as a rainfall and slope dataset,
must be discretely processed by a certain method. In this study, we divided the rainfall, slope and NDVI
into nine sections using the natural break method. The land use dataset (CCI LC) was reclassified into
nine categories based on the classification scheme of Table S1. We selected 816 randomly distributed
sample points with a spatial separation of at least 15 km as statistical units for model input, and the
distribution of sample points is listed in Figure S1. We conducted a geographical detector method with
ArcGIS     10.5     and     the     R     package     "geodetector"     (https://cran.r-
roject.org/web/packages/geodetector/index.html).
**3 Results**
**3.1 Distribution and monthly variation of soil erosion**

The soil erosion in the Yellow River basin in 2015 showed a high degree of spatial heterogeneity.

The areas with large amounts of soil erosion were mainly concentrated in the middle reaches of the
Yellow River. In Inner Mongolia, Shandong, southwestern Shaanxi, northern Ningxia and western Gansu,
the amount of soil erosion was small. There is a large risk of soil erosion in the eastern part of Qinghai,
southern Gansu, southern Ningxia and north-western Shaanxi, which is caused by pressures from soil
and water conservation.   From the perspective of the basin, the middle reaches of the Yellow River,
such as the Weihe River, face a high risk of soil erosion. Although the soil erosion intensity in the lower
reaches of the Yellow River is not high, the sediment caused by the erosion of the middle reaches of the
Yellow River causes sedimentation in the downstream riverbed, which further affects the atrophy and





uplift of the riverbed in the downstream area. The lower reaches of the Yellow River also face problems,
such as river channel siltation, reservoir lake siltation, and river bank erosion. Due to the thin soil layer
and the exposed rock in the area of Qinghai, although the current soil erosion intensity is low, the area
faces the potential danger of high soil erosion.
Figure 3 illustrates the boxplot of soil erosion and its scatter distribution for each month from 1995
to 2015. The amount of monthly soil erosion was significantly different from 1995 to 2015. The overall
numerical distribution showed a more pronounced symmetrical shape: the middle months were high, and
the values at the beginning and end of the year were lower. Specifically, soil erosion reached its highest
level in July with a multi-year average of 12.385. The average monthly soil erosion in the first and fourth
quarters was relatively low, at 2.006 and 3.332, respectively. Compared with March, the multi-phase soil
erosion in April increased by 115.79%. There was also a large drop in November compared with that in
October, with a decline of 57.81%. Furthermore, the soil erosion was extremely low in January and
December, with multi-phase averages of 0.833 and 0.526, respectively. However, the median amount of
multi-phase soil erosion in May was higher than that in June, but the average was slightly lower.
**3.2 Quantitative attribution analysis of yearly and monthly soil erosion distributions**
Figure 4 illustrates the quantitative attribution of soil erosion at the annual and monthly scales;
specifically, at the annual scale, topographic factors contribute more to soil erosion, while the dominant
factors in different time periods are different at the monthly scale. At the annual scale, the factors
affecting each factor did not change much and were relatively stable. From the annual scale, the slope
and surface roughness have a greater impact, while the rainfall and vegetation effects are ranked as three
or four. The topographical factor increased its influence before 2005, and the $q$ value reached values
above 0.2 and then experienced fluctuations in terms of its decline and rise. Because both are based on
DEM dataset generation, the effects of surface roughness and slope present a synergistic change. The
rainfall peaked in 2000, and the $q$ value followed with a small decline.
At the monthly scale, the shock of various influencing factors was very obviously, and rainfall and
slope factors had a greater impact at the beginning and end of the year, while in the middle of the year,
vegetation had a greater impact. Compared to the other months, the impacts of land cover in March are
the highest of those for the year. At the beginning and end of the year, when the rainfall and vegetation
coverage are relatively low, rainfall has a greater impact, while in periods of high rainfall and high



vegetation coverage, vegetation factors will play a leading role over the effects of other factors. The
spatial resolution of the NDVI dataset used in this study was 8 km and that of the land cover dataset was
300 m. The spatial resolution of the two was quite different, which caused the detailed land cover
information to be covered by the coarse-resolution vegetation information. Thus, the effect of land cover
on soil erosion would be underestimated in this study. In general, the contribution rate of a single factor
to soil erosion is low. Only in January 2005 did the q value of the rainfall impact reach 0.42, which was
the highest in the study. In other cases, the q value of the influencing factor of a single factor almost did
not exceed 0.3.

According to Figure 4, because there is some redundancy between slope and surface roughness and

the influence of land cover-related factors is low, the three main factors of topography, rainfall and
vegetation are selected for analysis. The effect of pairwise interactions among the three factors on soil
erosion was studied (Figure 5). In general, the interaction of two factors is more effective in explaining
soil erosion than is a single factor. Similarly, the annual scale suggests that the factors affecting each
factor change little and are relatively stable. At the monthly scale, the shock of various influencing factors
is very obvious.

From the annual scale, the synergy between the NDVI and slope plays a greater role, followed by

the synergy between the rainfall and slope. The q value of the two is approximately 0.4. The NDVI and
slope, the rainfall and slope, and the slope and vegetation are similar in several typical years, including
1995, 2000, 2005, 2010, and 2015. The q value showed an upward trend in 1995 – 2005, then decreased
slightly and finally increased. At the monthly scale, at the beginning and end of the year, the rainfall and
slope were synergistically dominant. In the middle of the year, the vegetation and slope factors were
dominant, and between 2000 and 2015, there were fewer time nodes that shared a combination of rainfall
and vegetation. The rainfall and slope factors showed a relatively obvious increase and then decreased,
reaching the lowest value around July. In several months, the synergy between rainfall and slope reached
its highest in January 1995, and its q value was 0.727. In July 2005, the lowest value was reached, and
its q value was 0.153. The synergy between vegetation and slope showed irregular oscillations in the
months of 1995 and 2000, while in 2005, 2010, and 2015, a certain peak was reached in the middle of
the year. The synergy between vegetation and rainfall presented irregular oscillations in the study years.





### 3.3 Quantitative attribution analysis of yearly and monthly soil erosion variability


Figure 6 shows the effect of annual and monthly scale single factors on soil erosion. At the annual
scale, the magnitude of the three factors is ranked as rainfall > slope > vegetation. In general, rainfall had
a higher impact on soil erosion than did the other two factors, and the trend of the effect of rainfall first
increased and then decreased. The impact reached its highest in 2005, with a q value of 0.287, and then
it experienced a decline, and the q value of rainfall in 2015 was less than 0.1. While the NDVI had a
small impact on soil erosion changes, it experienced a slow rise. The rainfall in 2015 experienced a large
increase compared to that in 2010.
At the monthly scale, the changes in the effects of the three factors are obvious, and the rainfall
factor tends to have a greater impact at the beginning and end of the year due to the obvious changes in
rainfall at the beginning and end of the year. The q value of the rainfall factor at the beginning and end
of the year is higher. In the middle of the year, the change of rainfall is relatively low, which results in a
lower impact on the amount of soil erosion in the adjacent months. For the vegetation factor, the time
period with the lowest impact of the whole year is the period with the smallest q value, which occurs
around July. Due to the year-round variation in the NDVI, the impact of vegetation on soil erosion
changes to a lower value in the middle of the year.
Figure 7 shows the contribution of the two-factor interactions to changes in soil erosion at annual
and monthly scales. At the annual scale, after 2005, the impact of the slope and rainfall interaction is
declining, but at all research nodes, the interaction of the slope and rainfall is the strongest among the
three factors, and the impact of vegetation on soil erosion rises. The interaction between the vegetation
and rainfall experienced an initial increase and then a decrease. At the monthly scale, the interaction
between the rainfall and slope presented a symmetrical pattern, with a greater effect at the beginning and
end of the year; furthermore, it reached its lowest value for the year around July. However, the others
showed a vibrating state. Overall, the two-factor interaction was more powerful than was the single-
factor interpretation, and changes in soil erosion were more sensitive to fluctuations in rainfall than to
fluctuations in vegetation.





**4 Discussion**

**4.1 Integrating temporal and spatial heterogeneity effects into soil erosion management**

Ecosystems are complex entities that span geographic and temporal scales and are inconsistent with various man-made jurisdictional and political demarcations (Bodin, 2017). Given these conditions, it is important for the structures of governance to solve the institutional fragmentation and match the temporal and spatial extents of ecosystem processes (Lubell, 2013). Cross-border and cross-scale collaboration is often seen here as a means by which to overcome such institutional fragmentation (Cosens, 2013; Walker et al., 2009). Therefore, it is urgent to integrate temporal and spatial heterogeneity effects into erosion management and to achieve a collaborative environmental governance framework for soil and water conservation.

According to Figure 3, soil erosion shows a high level of temporal variability, with soil erosion being highest in July and lower at the beginning and end of the year. The reason for this heterogeneity in soil erosion is because the parameters associated with soil erosion show an equally high spatial heterogeneity (Nearing et al., 1999). The period of the highest soil erosion during the year should be the period combined with high rainfall erosivity (high R factor) and low vegetation cover (high C factor). If the annual average data are used to blindly assess soil erosion on a detailed time scale, it may cause an incorrect estimate of soil erosion, which is not conducive to the implementation of soil and water conservation work.

Based on the analyses in Figures 4-7, we found that the distribution patterns of soil erosion and the factors that drive changes in soil erosion vary from month to month. In general, for this study area, rainfall has a greater impact during periods of low rainfall and vegetation coverage, while the contribution of vegetation is greater during periods of high vegetation coverage and rainfall. In short, we need to plan reasonable soil and water conservation work based on the characteristics of the time period. In recent years, demographic, cultural and political changes have had a strong impact on deforestation, replacing forests with croplands, and this practice has led to an increase in soil erosion (Begueria et al., 2006). A large range of soil and water conservation measures have been adapted to increase agricultural production and reduce soil erosion. These techniques are mainly concentrated on reducing slope correction/water velocity (i.e., bench terraces), increasing vegetation cover (i.e., cover crop, mulching, permanent cove with tree/crop/herbaceous associations and rangeland restoration) and/or improving soil




quality (i.e., amendments) (Raclot et al., 2018). However, these control measures become more
concentrated by changing the C factor or the LS factor. We found that the soil erosion distribution and
changes were more sensitive to the interaction of two factors compared to that of a single factor. In other
words, soil erosion control measures for two or more factors may have a significant improvement.
Furthermore, all of these techniques have been introduced with varying degrees of success depending on
the environmental and societal contexts (De Graaff et al., 2013; García-Ruiz et al., 2013).

The formulation and implementation of land use policies and ecological protection policies cannot

be constrained to certain administrative units (Chi and Ho, 2018). The management of soil erosion risk
should also break through the boundaries of administrative units; however, most work is based on the
three-level basin scale. Promoted by the Chinese Government, the River Chiefs system is well-placed to
coordinate various governmental departments and improve the efficiency and efficacy of a multitude of
water-resource management efforts, operating on the provincial, city, county, and township levels.
Drawing on the experience of the River Chiefs system, it is urgent to establish a water and soil
conservation management system based on different river basin level scales. Furthermore, human
behaviours and multiple ecosystem processes have been interconnected, and ecosystem management
may trigger possible unprecedented effects on the target and/or non-target processes (Zhao et al., 2018).
Therefore, soil and water conservation is by no means an isolated act because soil erosion control may
cause multiple effects from the local to regional scales (Fu et al., 2017). Using soil and water conservation
as a case study, there can be positive effects, such as soil conservation and C fixation, at the local scale
(Wang et al., 2015); however, it can also lead to environmental problems downstream, such as dried soil
layers and water shortages (Feng et al., 2016). Large-scale soil and water conservation requires cross-
sectoral and cross-regional trade-offs and coordination.
**4.2 The direction of model improvement**

Scale refers to the time and space dimension of the object of process under study, and the

appropriate scale for observations is a function of the type of environment and the type of information
desired (Woodcock and Strahler., 1987). The representation of geographical phenomena on the time and
space scales, as the time and space resolutions of observations change, the information that is obtained
also changes. The spatial scale of the application of RUSLE's original design should be only at the plot
scale. However, with the deepening of the research, the RUSLE model has been applied to larger scales,



e.g., nation (Van der kniff, 1999), continent (Van der kniff, 2000) and even global (Borrelli et all, 2013),
by adjusting the data sources, algorithms and parameters of some factors in RUSLE. However, the
exploration of using RUSLE at different temporal scales is still lacking, and a small number of studies
focus on the C factor for a more in-depth discussion (Alexandridis et al., 2015; Schmidt et al., 2018).
However, there has been a rapid advancement of remote sensing and GIS technology and an
improvement in the satellite revisiting cycle, which provides data with different spatial and temporal
resolutions and data downscaling methods. The data accumulated by long-term field testing also provide
extensive and accurate verification values for the validation and application of the model. Overall, a lack
of data is no longer a hindrance to the development of soil erosion models. High temporal resolution
products based on MODIS data series have been widely used. The high temporal resolution of soil
erosion mapping should also receive attention.
Based on the study of Figure 4 and Figure 6, slope has a greater impact on the spatial distribution
of soil erosion, and the change in soil erosion is more sensitive to the change in rainfall. The finer R
factor method and rainfall datasets can more accurately characterize the change in soil erosion, while the
finer LS factor and method can invert the spatial distribution of soil erosion. Of course, any improvement
in data, method, and parameters for each factor in the RUSLE model can effectively improve the
accuracy of soil erosion, but it may be a more efficient direction to explore the R or LS factors in depth
over the other factors.
Many of the currently developed C factor formulas combine land use and NDVI data (Panagos et
al., 2015; Jiang et al., 1996; Liu et al, 2010). However, the inconsistency of the spatial resolution scale
of the NDVI and land cover data result in greater uncertainty of the research in specific applications.
Therefore, the adaptability of the spatial resolution of the two kinds of data should be fully considered in
the development of C factor formulas that combine vegetation and land cover data.
**4.3 Uncertainty analysis and future perspectives**
The method used to evaluate the factors affecting soil erosion was the geographical detector method,
but the input of independent variable data used by this tool must be discretized according to certain
principles. The choice of discretization methods will inevitably affect the interpretation of the final results.
According to the previous experience of soil erosion (Gao and Wang, 2019), we used the natural break





method, and the input data were divided into 9 categories. Other classification methods, such as the
geometrical interval and equal interval methods, are also worth trying.

This study applies the RUSLE model to a monthly scale, which violates the original intention of

the RUSLE model design, but we think this was an effective attempt. The amount of monthly scale
erosion that may be assessed is not accurate but reflects the trend in soil erosion at a monthly scale to
some extent. We believe that this study provides many useful ideas and inspirations for soil erosion
assessment and control.
**5 Conclusion**

The current study identified the dominant factors (and combinations of factors) of soil erosion in

the Yellow River basin of China and its variability in the typical years of 1990, 1995, 2000, 2005, 2010
and 2015 based on the RUSLE model and the geographical detector method.

Topographical factors such as slope and surface roughness have a greater impact on the spatial

distribution of soil erosion, while rainfall and vegetation are as follows. In periods of low rainfall and
vegetation coverage, the interaction of rainfall and slope has a great influence on the distribution of soil
erosion, while in periods of high vegetation coverage and high rainfall, the spatial distribution of soil
erosion is greatly affected by the synergy of vegetation and slope. The change in rainfall contributes
greatly to the change in soil erosion, but the contribution decreases each year, and the contribution of
vegetation change increases each year.

We found that the distribution patterns of soil erosion and the factors that drive changes in soil

erosion vary from month to month and vary from area to area. It is necessary to combine the temporal
and spatial heterogeneity with the soil erosion management and form a collaborative environmental
governance framework. A finer LS factor formula, terrain datasets, R factor formula and rainfall datasets
can more accurately characterize the distribution and variation of soil erosion. Future research needs to
develop soil erosion assessment models for higher temporal resolutions (monthly scale) to cope with soil
erosion risks.





**Acknowledgements**

This research was funded by the National Key R&D Program of China (No. 2017YFA0604704),

the National Key Research Program of China (No. 2016YFC0501604), and the State Key Laboratory of
Earth Surface Processes and Resource Ecology (No. 2017-FX-01(2)).

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





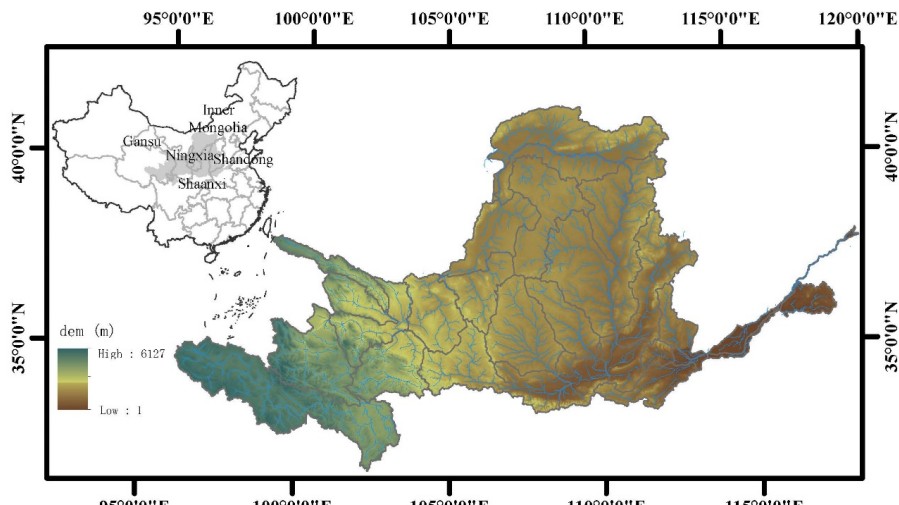


**Figure 1: The location of the study area in China and the regional topography.**

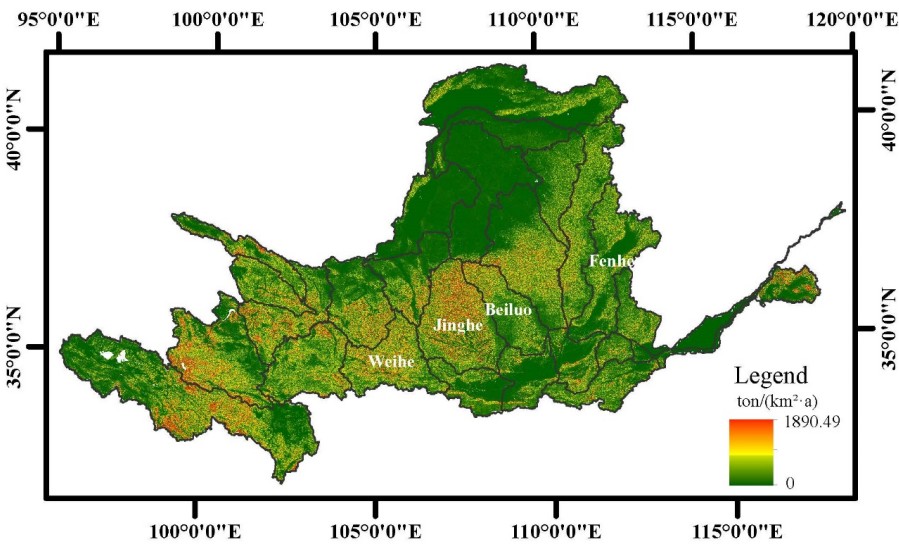


**Figure 2: Distribution of soil erosion in the Yellow River basin in 2015.**


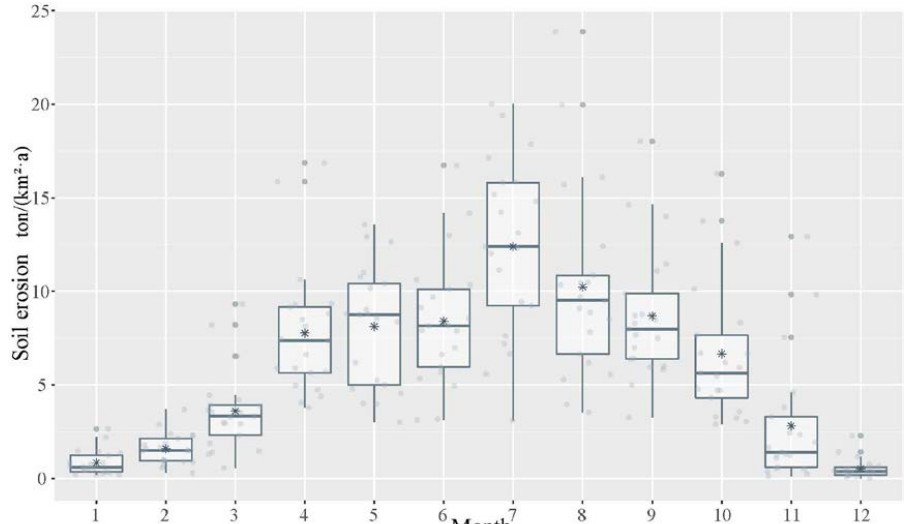

**Figure 3: Variation in average monthly soil erosion from 1995 to 2015.**

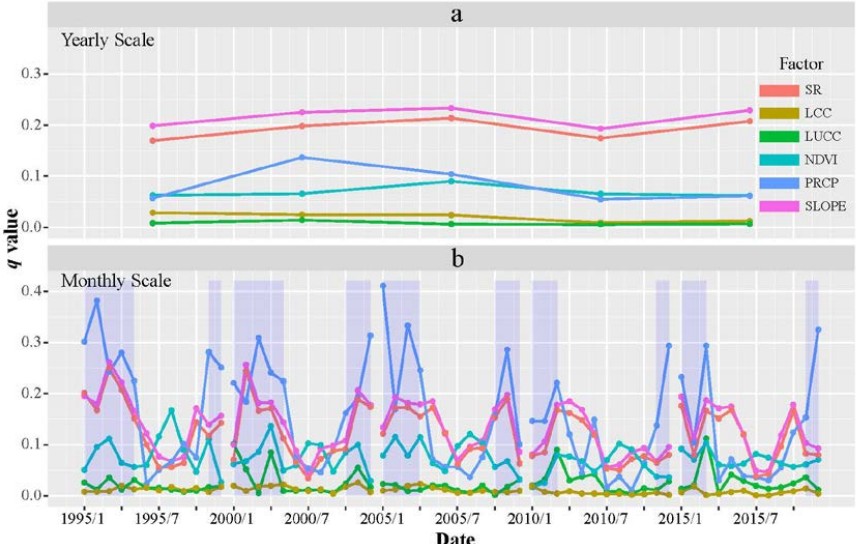

**Figure 4: Contribution analysis of a single factor to the soil erosion distribution on a yearly and monthly scale.**
**SR refers to the surface roughness, LCC refers to the land cover complexity, LUCC refers to the land use and**
**land cover change, NDVI refers to the normalized difference vegetation index, PRCP refers to the**
**precipitation and SLOPE refers to the surface slope gradient.**





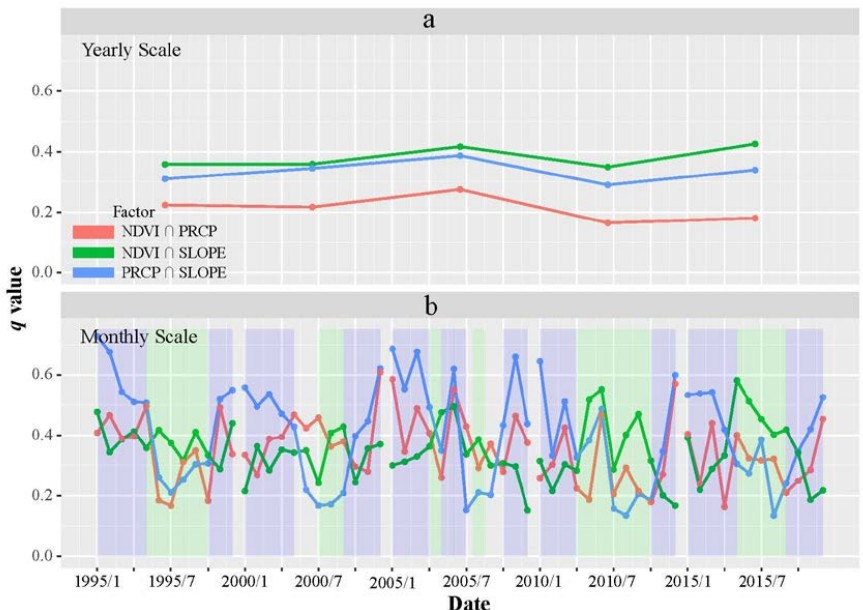


**Figure 5: Contribution analysis of multiple interacting factors to soil erosion distribution on a yearly and monthly scale, where NDVI refers to the normalized difference vegetation index, PRCP refers to the precipitation and SLOPE refers to the surface slope gradient.**

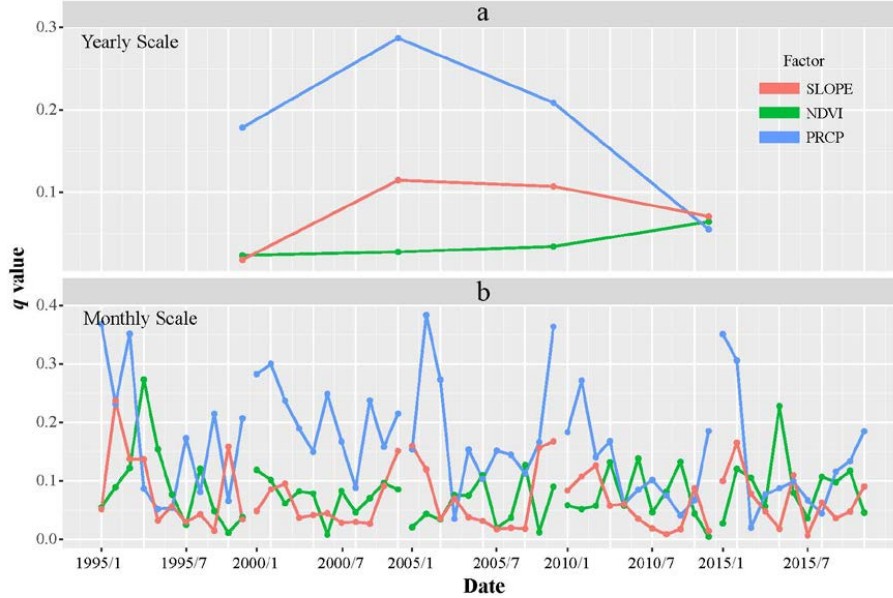

**Figure 6: Contribution analysis of a single factor to soil erosion variability on a yearly and monthly scale, where NDVI refers to the normalized difference vegetation index, PRCP refers to the precipitation and SLOPE refers to the surface slope gradient.**





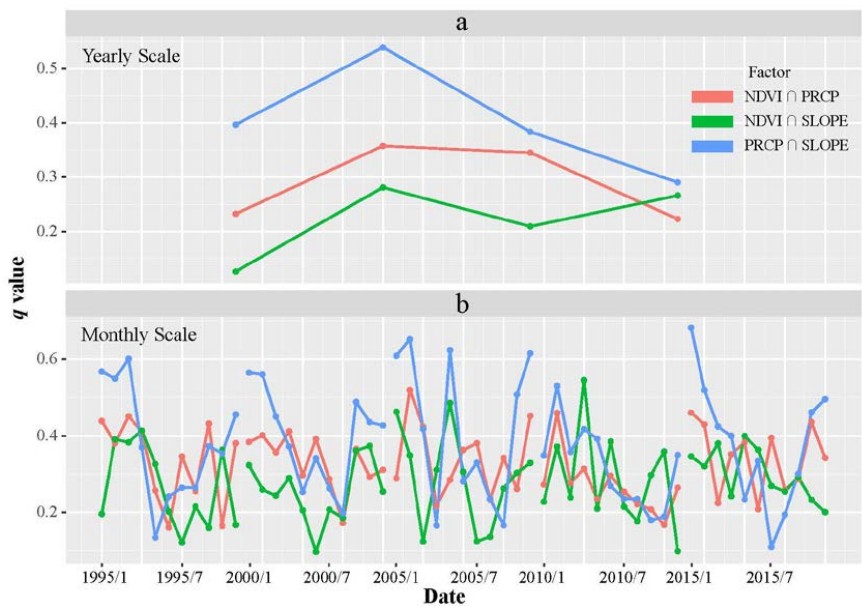


**Figure 7**: **Contribution analysis of multiple interacting factors to soil erosion variability in yearly**

**and monthly scales, where NDVI refers to the normalized difference vegetation index, PRCP refers to the**
**precipitation and SLOPE refers to the surface slope gradient.**