# Peer review of "Influencing factors and their interactions of water erosion"

_Natural Hazards and Earth System Sciences, 2019_

## Short Comment (SC1) · 14 May 2019

Soil erosion in the Yellow River Basin is a serious natural disaster. The authors used the RUSLE model to assess the risk of soil erosion and quantitatively identified the impact factors of soil erosion. Compared to previous studies, the authors attempted to quantify the multifactorial interactions with soil erosion based on geographical detector method. The logic consistency and method selection are ok, while, it lack the explanation and comparisons with other studies in Discussion Section. If the author can modify it according to the following comments, the content and structure of the article may be more abundant and complete. Therefore, I suggest this paper to be accepted

after revisions.

1. Introduction Lines 38-54 There is a problem with the connection between the sentences, and further revisions are needed. In particularly, it is necessary to further supplement the work of scholars in studying the impact factors of soil erosion.

2. Lines 94,99 Reference should be supplemented.

3. Table S2: Need to be explain the meaning of some parameters in the table such as BS, BW.

4. 2.2 Data and processing The geographical detector method was introduced, but there is a lack of expression on how to apply the geographical detector method specifically to this study. Please provide additional explanation.

5. Lines 198-207: Need to supplement the unit for the result.

6. Result: For some of the results, consider whether to use the moving average method to reprocess results to reduce some change factors and the uncertainty of the results.

7. Discussion: Soil erosion in the Yellow River Basin (Loess Plateau) is a hot topic of research. Authors need to compare their soil erosion assessment results with previous results.

8. Discussion: The author seems to be comparing the effects of two single factors and the effects of interactions, and whether the interaction between the two factors is enhanced or weakened compared to the original two separate effects.

9. Discussion 4.2 The direction of model improvement: The author needs to summarize what improvements have been made to the RUSLE model by scholars in the past literature, and then combine authors' research results to point out the potential model improvement direction.

2019-122, 2019.

---

## Referee Comment (RC1) · Anonymous Referee #1 · 30 May 2019

The author assessed water erosion risk in the Yellow River Basin and highlighted the impact of interaction of the environment factors on soil erosion. This is a nice research on the relationship between soil erosion and environment factors. While, this is a lack of comparison about the effects of a single environmental factor and multiple environmental factors. In addition, there are some problems in the details of figures and texts. Therefore, I recommend this manuscript to be accepted after revisions.

Introduction: I noticed that the topic of this special issue is Remote sensing, modelling-based hazard and risk assessment, and management of agro-forested ecosystems. Soil erosion is indeed disaster and the authors have used modelling methods in soil

erosion assessments. However, the article lacks the expression of the relationship between soil erosion and the agro-forestry ecosystems. The author needs to supplement the relationship in the Introduction and Disccusion.

Introduction: Lines 42-46: The author attempts to explain the complex relationship between soil erosion and environmental factors using a few examples, but the expression here is too fragmented. Author needs to summarize the meanings expressed by these sentences in a general language.

Lines 198-206: The results in these sentences are missing units. Pleas further indicate the unit.

Figure 3: Need to explain some symbols in the box plot.

Figure4: Purple and green shadows appear in these figures such figure4 and figure 6, and the author need to explain these shadows.

Discussion: The author carried our research on the interaction of environmental factors on soil erosion, but need to further supplement the comparison between the interaction of multiple factors and effect of single factor.

Discussionïij Ž4.3: Need to supplement the impact of the missing P factor calculation on uncertainty of results.

---

## Referee Comment (RC2) · Anonymous Referee #2 · 30 May 2019

Soil erosion is a global ecological environment problem that limits the sustainable development of social economy, the study of soil erosion in the Yellow River Basin is of significant importance for livelihood of local resident. However, this study didn't give the meaningful contribution to soil erosion control. The comments and suggestions are as follows: (1) What is the meaning to do the quantitative attribution analysis of soil erosion in different months, as we all know, in the rainy season soil erosion is much higher than other months. In addition, the policy made to control soil erosion is usually based on the spatial distribution of soil erosion, not based on temporal distribution of soil erosion. For example, the construction of check dam, the dam will be used for a long time, rather than during some months. (2) The quantitative attribution analysis of soil erosion

and its variability have been done by other researchers previously, thus innovation was lacked in this paper. (3) In page 2, line 51, there is a problem in the format of reference cited. (4) Please check the unit of soil erosion calculated by RUSLE model, ton/(km2 a) or ton/(ha a). (5) Do you have any verification of the results simulated by RUSLE model? The validity of the results is the basement for the following analysis. By the way, I doubt the soil erosion module mentioned in abstract, it is too small for the Yellow River Basin. (6) In the uncertainty analysis part," Other classification methods, such as the geometrical interval and equal interval methods, are also worth trying", why don't you try other methods and select the most appropriate method for your study. (7) The language expression need to be improved, for example, "Topographical factors such as slope and surface roughness have a greater impact on the spatial distribution of soil erosion, while rainfall and vegetation are as follows." In this language, there is no adversative relation.

---

## Author Comment (AC1) · 7 Aug 2019

1. Introduction: I noticed that the topic of this special issue is Remote sensing, modelling based hazard and risk assessment, and management of agro-forested ecosystems. Soil erosion is indeed disaster and the authors have used modelling methods in soil erosion assessments. However, the article lacks the expression of the relationship between soil erosion and the agro-forestry ecosystems. The author needs to supplement the relationship in the Introduction and Discussion.

Answer: Thank you very much for your suggestion. To better match the topic of this special issue, we have added some statement about the relationship between soil

erosion and the agro-forestry ecosystems (please see Lines 32-37, 338-345 of the manuscript).

2. Introduction: Lines 42-46: The author attempts to explain the complex relationship between soil erosion and environmental factors using a few examples, but the expression here is too fragmented. Author needs to summarize the meanings expressed by these sentences in a general language.

Answer: Thank you very much for the opinion of the reviewer, we have modified the original sentence in order to better express and these sentences have been revised into "The identification of the mechanisms of soil erosion and factors affecting soil erosion is an important basis for land use management and ecosystem government. Several studies have focused on determining the driving forces affecting soil erosion, including precipitation, geomorphology, land use type, vegetation, and soil physical properties (Vrieling, 2006; Zhou et al., 2008; Peng and Wang, 2012; Gao and Wang., 2018; Beskow et al., 2009; Tian et al., 2009). Climate factors such as precipitation, temperature and evaporation, all affect regional soil erosion. Among them, rainfall is one of the most important factors affecting erosion. From the time of rainfall exposure to the land surface, the process of splashing and spurting by raindrops has a significant impact on soil erosion. Terrain is one of the important natural geomorphic factors affecting soil erosion, and it is one of the lower interface factors affecting the formation and development of soil erosion. Different characteristics of topography and their changing tendency correspond to different features of slope runoff and confluence, which affect the occurrence and intensity of soil erosion directly (Yang et al., 2007). Among them, the influence of slope on soil erosion is finally reflected by the runoff of the slope and their flow velocity, which is an important factor restricting the spatial distribution of productivity. For vegetation, the vegetation canopy can protect the surface soil from direct impact from raindrops and weaken runoff, thus eventually reducing soil erosion. Excessive land reclamation, unreasonable production activities and land use patterns, and the reductions in surface vegetation cover have a magnifying effect on soil erosion (Wu

and Cai., 2003)" (please see Lines 43-60 of the manuscript).

3. Lines 198-206: The results in these sentences are missing units. Pleas further indicate the unit.

Answer: We have already added these units and please see Lines 221-231 of the manuscript.

4. Figure 3: Need to explain some symbols in the box plot.

Answer: Thank you for your reminder and we have already explained these symbols in Figure 3. The solid point represents the amount of soil erosion during this time period, the asterisk represents the average amount of soil erosion during this period, and the horizontal line refers to the median of soil erosion during this period (please see Lines 626-628 of the manuscript).

5. Figure 4: Purple and green shadows appear in these figures such figure4 and figure 6, and the author need to explain these shadows.

Answer: Thank you for your reminder, we have already added it in the corresponding place. In order to express the specific meaning more intuitively, we used different colours of shadows. The shadows of different colours show that the factors represented by the colour during this time period, which contribute more to soil erosion than other factors. Please see Lines 633-635, 639-641 of the manuscript.

6. Discussion: The author carried our research on the interaction of environmental factors on soil erosion, but need to further supplement the comparison between the interaction of multiple factors and effect of single factor.

Answer: Thank you for your helpful suggestion and we have added the expression and 4 figures about the comparison between the interaction of two factors and effect of single factor (please see Lines 275-281, 308-315 of the manuscript, and Figure S2-S5 of the supplement).

7. Discussion4.3: Need to supplement the impact of the missing P factor calculation on uncertainty of results.

Answer: Thank you for suggestion and we have added some discussion about the uncertainty of the impact of the missing P factor calculation (Please see Lines 415-421 of the manuscript).

Please also note the supplement to this comment:
https://www.nat-hazards-earth-syst-sci-discuss.net/nhess-2019-122/nhess-2019-122-AC1-supplement.zip

---

## Author Comment (AC2) · 7 Aug 2019

1. What is the meaning to do the quantitative attribution analysis of soil erosion in different months, as we all know, in the rainy season soil erosion is much higher than other months. In addition, the policy made to control soil erosion is usually based on the spatial distribution of soil erosion, not based on temporal distribution of soil erosion. For example, the construction of check dam, the dam will be used for a long time, rather than during some months.

Answer: Thank you for your helpful comment and the reasons for my analysis on the monthly scale are as follows 1) For the RUSLE model, the soil erodibility (K factor)

and topography (LS) factors are stable over a long time period and are relatively in-dependent of anthropogenic interventions. However, the rainfall erodibility (R factor) and vegetation cover and management factor (C factor) are seasonally variable. The amount of soil erosion are seasonally variable. More and more people are paying at-tention to the study of soil erosion at the monthly scale. Some studies began to focus on the C factor for a more in-depth discussion at the monthly scale, such as the calcula-tion method of C factor, the characteristics of C factor in different periods (Alexandridis et al., 2015; Schmidt et al., 2018). In summary, the discussion of seasonal changes in soil erosion is necessary. a. Alexandridis, T. K., Sotiropoulou, A. M., Bilas, G., Karapet-sas, N., and Silleos, N. G.: The Effects of Seasonality in Estimating the C-Factor of Soil Erosion Studies, Land Degradation & Development, 26, 596-603, 2015. b. Schmidt, S., Alewell, C., and Meusburger, K.: Mapping spatio-temporal dynamics of the cover and management factor (C-factor) for grasslands in Switzerland, Remote Sensing of Environment, 211, 89-104, 2018. 2) We strongly agree with your point of view that the construction of check dam, the dam will be used for a long time, rather than dur-ing some months. However, the policy is not only a long-term engineering measure, but also includes early warning of soil erosion disasters by various government de-partments. The focus of our research is not just on the monthly scale of soil erosion assessment, but on the contribution of different environmental factors and the interac-tion of two environmental factors to the spatial distribution and spatial variation of soil erosion in different months. We have come up with many interesting conclusions that in most cases the two-factor interaction to the spatial distribution and spatial variation of soil erosion exhibits a nonlinearly enhanced state. From this perspective, the research of soil erosion on a monthly scale is important.

2. The quantitative attribution analysis of soil erosion and its variability have been done by other researchers previously, thus innovation was lacked in this paper.

Answer: Thank you very much for allowing us to reconsider the innovation of this manuscript. While, it seems that few people have studied the interaction of two environmental factors on soil erosion. We applied an interaction detector module of geographical detector to the research of distribution and spatial variation of soil erosion. And we used four figures and a lot of text to elaborate on this work. We found that the two-factor interaction exhibits a nonlinearly enhanced state in most cases (please see Lines 275-281, 308-315 of the manuscript, and Lines 4-16 of the supplement). In summary, we believe that our work is still innovative.

3. In page 2, line 51, there is a problem in the format of reference cited.

Answer: We apologize for our overlook and have revised the format of reference cited (please see Lines 68-71 of the manuscript).

4. Please check the unit of soil erosion calculated by RUSLE model, ton/(km2 a) or ton/(ha a).

Answer: We apologize for our overlook and have revised these units (please see Lines 22, 225-231 of the manuscript).

5. Do you have any verification of the results simulated by RUSLE model? The validity of the results is the basement for the following analysis. By the way, I doubt the soil erosion module mentioned in abstract, it is too small for the Yellow River Basin.

Answer: Thank you very much for your helpful comments. The RUSLE model is an effective tool for rapid assessment of soil erosion. And using the detailed surface information provided by remote sensing, the RUSLE model has successfully been applied to a variety of spatial scale assessments of soil erosion, from the plot scale to the global scale (Thiam, 2003; Vrieling, 2006; Van der kniff, 1999; Van der kniff, 2000; Borrelli et al., 2013). We think the result of soil erosion is reliable. The values appearing in the Abstract are only the amount of soil erosion we estimated in July based on the RUSLE model. It is not the amount of soil erosion throughout the year, so it may be much smaller than the impression. In addition, considering that our research mainly focuses on the impact of environmental factors on soil erosion, the data input required

by Geographical detector needs to be reclassified. In other words, we need to ensure that the relative value of the results is accurate, and there is no strict requirement for the absolute value of the data and the evaluation results.

6. In the uncertainty analysis part," Other classification methods, such as the geometrical interval and equal interval methods, are also worth trying", why don't you try other methods and select the most appropriate method for your study.

Answer: Thank you for your suggestion. According to the paper of the inventor of geographical detector, it is emphasized that a variety of classification methods are applicable to geographical detector. In other words, geographical detector are not mandatory for which classification method to use. We referred to a paper on the application of geographical detector to soil erosion assessment, and we decided to use the natural break method in this study. In addition, in order to avoid a certain degree of misinformation, we have added a description in the 2.2 Data and processing (Please see 194-197), and removed the original sentence in 4.3 Uncertainty analysis and future perspectives.

7. The language expression need to be improved, for example, "Topographical factors such as slope and surface roughness have a greater impact on the spatial distribution of soil erosion, while rainfall and vegetation are as follows." In this language, there is no adversative relation.

Answer: Thanks to reviewers for pointing out grammatical errors in manuscript, we have modified similar expressions (Please see Lines 14-16, 431-432). In addition, we went over this paper carefully and invited a native speaker of English from American Journal Experts (https://www.aje.com/) to improve the readability of our paper.

Please also note the supplement to this comment:
https://www.nat-hazards-earth-syst-sci-discuss.net/nhess-2019-122/nhess-2019-122-AC2-supplement.zip

2019-122, 2019.

---

## Author Comment (AC3) · 7 Aug 2019

1. Introduction Lines 38-54 There is a problem with the connection between the sentences, and further revisions are needed. In particularly, it is necessary to further supplement the work of scholars in studying the impact factors of soil erosion.

Answer: We have modified the original sentences to make the logic of the statement more coherent. Besides that, we added some expression about the work of scholars in studying the impact factors of soil erosion, especially rainfall, topography and vegetation (please see Lines 43-60 of the manuscript).

[Figure]

2. Lines 94,99 Reference should be supplemented.

Answer: We have already supplemented the references (please see Lines 110-116 of the manuscript).

3. Table S2: Need to be explain the meaning of some parameters in the table such as BS, BW.

Answer: Thank you for your comments. These abbreviations represent different kÓğp-pen climate zones. We have already added a note to Table S2 (please see Lines 20-22 of the supplement).

4. 2.2 Data and processing. The geographical detector method was introduced, but there is a lack of expression on how to apply the geographical detector method specifically to this study. Please provide additional explanation.

Answer: Thank you very much for pointing out the shortcomings of our manuscript. For the overall integrity of the manuscript, we have added some expression on how to apply the geographical detector method to this work (please see Lines 200-204 of the manuscript).

5. Lines 198-207: Need to supplement the unit for the result.

Answer: We have already supplemented these units (please see Lines 221-231 of the manuscript).

6. Result: For some of the results, consider whether to use the moving average method to reprocess results to reduce some change factors and the uncertainty of the results.

Answer: Thank you for your suggestion. For long-term continuous sequence data, the moving average method has the advantage of eliminating occasional fluctuations. However, specific to this study, we studied the monthly scale study of five years (1995, 2000, 2005, 2010, 2015), the essence of which is still not continuous data of long time series. Accordingly, the recursive operation in the moving average method is not well

applied to the monthly scale study of this work. In summary, we did not use the moving average method.

7. Discussion: Soil erosion in the Yellow River Basin (Loess Plateau) is a hot topic of research. Authors need to compare their soil erosion assessment results with previous results.

Answer: We searched the literature in several databases and found that there is less research on soil erosion in the whole Yellow River Basin as the study area, mostly concentrated in a small basin within the Yellow River Basin or the Loess Plateau. Considering the applicability of the scale of the data and methods, this makes the comparison of the result of soil erosion difficult. We finally chose to compare the results of a Chinese literature study (Li et al., 2010). The relative values of the spatial distribution of soil erosion are consistent with our calculations. Considering that our research focuses on the impact of environmental factors on soil erosion, the data input required by Geographical detector needs to be reclassified. In other words, we need to ensure that the relative value of the results is accurate, and there is no strict requirement for the absolute value of the data and the assessment results. In addition, the RUSLE model is an effective tool for rapid assessment of soil erosion. And using the detailed surface information provided by remote sensing, the RUSLE model has successfully been applied to a variety of spatial scale assessments of soil erosion, from the plot scale to the global scale (Thiam, 2003; Vrieling, 2006; Van der kniff, 1999; Van der kniff, 2000; Borrelli et al., 2013). We think the result of soil erosion is reliable.

8. Discussion: The author seems to be comparing the effects of two single factors and the effects of interactions, and whether the interaction between the two factors is enhanced or weakened compared to the original two separate effects.

Answer: Thank you for your helpful suggestion and we have added the expression and 4 figures about the comparison between the interaction of two factors and effect of single factor (please see Lines 275-281, 308-315 of the manuscript, and Lines 4-16 of

the supplement). And we found that in most cases the two-factor interaction exhibits a nonlinearly enhanced state.

9. Discussion 4.2 The direction of model improvement: The author needs to summarize what improvements have been made to the RUSLE model by scholars in the past literature, and then combine authors' research results to point out the potential model improvement direction. Answer: Thank you for your helpful suggestion and we have added the expression about what improvements have been made to RUSLE model by scholars, especially the correlation of factors (please see Lines 385-398, 400-404 of the manuscript).

Please also note the supplement to this comment:
https://www.nat-hazards-earth-syst-sci-discuss.net/nhess-2019-122/nhess-2019-122-AC3-supplement.zip